# Microvascular Contributions to Alzheimer Disease Pathogenesis: Is Alzheimer Disease Primarily an Endotheliopathy?

**DOI:** 10.3390/biom13050830

**Published:** 2023-05-13

**Authors:** Rawan Tarawneh

**Affiliations:** Department of Neurology, Center for Memory and Aging, University of New Mexico, Albuquerque, NM 87106, USA; raltarawneh@salud.unm.edu; Tel.: +1-505-272-5429

**Keywords:** endothelium, pericyte, capillary, neurodegeneration, Alzheimer disease

## Abstract

Alzheimer disease (AD) models are based on the notion that abnormal protein aggregation is the primary event in AD, which begins a decade or longer prior to symptom onset, and culminates in neurodegeneration; however, emerging evidence from animal and clinical studies suggests that reduced blood flow due to capillary loss and endothelial dysfunction are early and primary events in AD pathogenesis, which may precede amyloid and tau aggregation, and contribute to neuronal and synaptic injury via direct and indirect mechanisms. Recent data from clinical studies suggests that endothelial dysfunction is closely associated with cognitive outcomes in AD and that therapeutic strategies which promote endothelial repair in early AD may offer a potential opportunity to prevent or slow disease progression. This review examines evidence from clinical, imaging, neuropathological, and animal studies supporting vascular contributions to the onset and progression of AD pathology. Together, these observations support the notion that the onset of AD may be primarily influenced by vascular, rather than neurodegenerative, mechanisms and emphasize the importance of further investigations into the vascular hypothesis of AD.

## 1. Introduction

Alzheimer disease (AD) is the most common cause of dementia, accounting for over 70% of dementia cases in individuals above the age of 65 years [1]. The two main pathological hallmarks of AD are extracellular deposits of the amyloid-β (Aβ) protein in the form of amyloid plaques and intracellular aggregates of hyperphosphorylated tau protein in the form of neurofibrillary tangles [2]. Current disease models are based on the notion that abnormal protein aggregation is the primary event in AD, which begins a decade or longer prior to symptom onset, and ultimately leads to synaptic injury and neurodegeneration [3].

Vascular disease, including arteriolosclerosis, atherosclerosis, microinfarcts, and cerebral amyloid angiopathy (CAA), is a common co-pathology which is observed in 20–80% of AD brains at autopsy [4,5,6,7]. Furthermore, almost all AD brains display evidence of endothelial and capillary degeneration even in the absence of other forms of macrovascular pathology [8,9,10,11,12]. Significant and bidirectional interactions between AD and various forms of vascular pathology have been well documented [13]; amyloid and tau toxicity disrupts the blood–brain barrier (BBB) and alters vascular permeability, and structural or functional damage to cerebral vasculature impairs amyloid clearance and promotes tau aggregation. Previous neuropathological studies examining vascular pathology in AD have focused primarily on pathology within the small- and medium-sized arteries and arterioles; however, there is growing evidence to suggest that “micro”-vascular disease (i.e., at the capillary level) and alterations to specific vascular constituents, such as endothelium and pericytes, play an important role in AD pathogenesis [8,9,14,15,16,17,18]. From a functional standpoint, cross-sectional and longitudinal clinical studies and animal models that utilize advanced vascular imaging methods suggest that impaired cerebral blood flow (CBF) is a common and early predictor of AD pathology, which possibly precedes abnormal protein aggregation and directly contributes to neuronal and synaptic loss in even the earliest pre-symptomatic stages of the disease [8,19].

This review will examine evidence from clinical, imaging, neuropathological, and animal studies supporting vascular contributions to the onset and progression of AD pathology, including amyloid and tau aggregation, as well as direct vascular contributions to synaptic injury, impaired axonal repair, and neurodegeneration. Importantly, the present review discusses evidence supporting the role of specific vascular constituents (e.g., pericytes, endothelium, astrocytes, and basement membrane proteins) and vascular abnormalities at the capillary level (i.e., “microvascular” disease) in mediating AD onset and progression. Together, these observations support the notion that the onset of AD may be primarily influenced by vascular, rather than neurodegenerative, mechanisms and emphasize the importance of further investigations into the vascular hypothesis of AD.

## 2. Evidence to Support Vascular Contributions to AD Pathogenesis

Several lines of evidence support the presence of vascular contributions to AD pathogenesis including associations of vascular pathology (e.g., arteriolosclerosis, microinfarcts, and CAA) with mechanisms involved in AD pathogenesis, including amyloid and tau aggregation, oxidative stress, mitochondrial dysfunction, and vascular inflammation, which cumulatively contribute to synaptic loss and neurodegeneration [8,9,13,14,15,19,20]. Importantly, there is recent evidence to suggest associations of specific vascular constituents (e.g., endothelium or pericytes) with molecular mechanisms or pathways directly involved in synaptic plasticity, axonal growth, neuronal repair, and memory or learning functions (e.g., long-term potentiation) [21,22].

The vascular hypothesis of AD, which was first proposed over 30 years ago, is based on the notion that the first inciting event in AD pathogenesis is vascular dysregulation, which then initiates a cascade of molecular and neuropathological changes leading to neuronal dysfunction and the onset of AD pathologies, such as amyloid and tau aggregation [23,24,25,26]. A revised version of the vascular hypothesis suggests a “two-hit” process, in which amyloid-independent microvascular damage (hit one) occurs due to several mechanisms (described below), resulting in impaired amyloid clearance and then subsequent amyloid toxicity (hit two) exacerbates microvascular injury via a positive feedback loop [20]. In various versions of the vascular hypothesis, the inciting event in AD onset is vascular dysregulation and reduced CBF, which precedes, causes, and then further exacerbates protein aggregation and neuronal or synaptic injury, the latter driving the cognitive and behavioral decline observed in the symptomatic stages [8].

This review summarizes the literature supporting this hypothesis and presents new data suggesting additional direct links between specific vascular components and cognition. Importantly, this current review expands the vascular hypothesis beyond associations of macrovascular pathology with AD to highlight the importance of “micro”-vascular constituents at the cellular and molecular level in AD onset and progression.

## 3. Vascular Risk Factors Increase the Risk for AD

The association of vascular risk factors, including hypertension, hyperlipidemia, diabetes mellitus (DM), obesity, and metabolic syndrome, with AD has been well documented in several clinical and epidemiological studies [8,27,28,29,30,31,32,33]. Low insulin levels [34] and DM [35] have been linked to AD risk in several other large studies, with hazard ratios of 1.31 and 1.62, respectively. A systematic review of 15 epidemiological studies reported a pooled adjusted hazard ratio risk of 1.57 with DM [36,37]. Increased AD risk in individuals with DM was found to be almost comparable to, and synergistic with, that of the Apolipoprotein E4 (*APOE4)* genotype, the most significant genetic risk factor for AD [35,38]. Apolipoprotein E, a major cholesterol transporter in the brain, has three isoforms: *E2*, *E3*, and *E4*. The presence of 1 or 2 *APOE4* alleles increases AD risk by 2–3-fold and 10–15-fold, respectively, due to increased amyloid and tau aggregation, dysregulated cholesterol metabolism, and impaired synaptic plasticity [39,40,41]. Carriers of the *APOE4* allele have a higher risk for both late-onset AD and cardiovascular disease, further supporting an important link between vascular disease and AD pathogenesis [42].

Other studies have shown associations of AD with midlife hypertension, while associations of AD with late-life hypertension have been inconsistent. In the longitudinal Honolulu-Asia Aging Study, midlife hypertension was associated with a higher risk for AD in middle-aged Japanese American men followed over 25 years, who demonstrated neuropathological evidence of increased hippocampal tau pathology and brain atrophy over follow-up, compared to normotensive controls [43]. Interestingly, midlife hypertension increased the risk of mild cognitive impairment (MCI) to a similar degree as *APOE4* [44,45,46]. AD risk increases with higher levels of low-density-lipoprotein (LDL) cholesterol and lower levels of high-density-lipoprotein (HDL) cholesterol [47,48]. Further, early onset AD has been linked to coding variants in *APOB*, which codes for apolipoprotein B, an important component of LDL [49]. High LDL has been linked to AD pathogenesis by increasing the risk for cerebrovascular disease, vascular inflammation, increased Aβ synthesis, Aβ-mediated synaptic toxicity, and tau aggregation [50,51,52].

The Rotterdam study, which examined over 7000 older adults, reported a higher risk for AD in association with vascular risk factors, such as DM, thrombotic episodes, high fibrinogen or homocysteine levels, atrial fibrillation, smoking, alcoholism, and atherosclerosis [8,53]. Participants with severe atherosclerosis had a three-times higher risk of developing AD or vascular dementia [53]. In the large longitudinal Atherosclerosis Risk in Communities (ARIC) study, which followed a multiracial cohort of middle-aged adults for 25 years, vascular risk factors, such as midlife smoking, diabetes, prehypertension, and hypertension, were associated with a higher risk for dementia including AD [38]. Elevated homocysteine levels, which are independent risk factors for vascular disease, are also associated with a higher risk for AD, brain atrophy, and tau pathology [54,55]. High homocysteine has been linked to impaired post-synaptic GABA (γ-aminobutyric acid) signaling, matrix metalloproteinase-9 (MMP-9) activation, and increased oxidative injury, which disrupts the BBB and contributes to AD pathogenesis [56].

Furthermore, vascular risk factors appear to have additive or synergistic effects on AD risk, with a higher risk being reported in the presence of multiple vascular risk factors [57,58]. One study found that diabetes and current smoking were the strongest risk factors in isolation or in clusters, while hypertension and heart disease were related to a higher risk of AD when clustered together or with either diabetes or smoking [59].

## 4. Concomitant Macrovascular Disease Is Present in Most, and Microvascular Disease in Almost All, AD Brains

Vascular disease, including subcortical white matter lesions or lacunes, is a common co-pathology in AD, being reported in 18–84% of AD brains [5,7]; however, the degree to which these forms of vascular disease contribute to neurodegeneration is yet to be determined. Despite their co-occurrence, AD and vascular pathology often do not localize in the same brain regions [5]. Furthermore, it remains unclear whether the severity of vascular disease correlates with the severity of neuronal or synaptic loss in AD brains [5]. While several studies from various cohorts have shown that white matter hyperintensities (WMH), which are used as an imaging surrogate of small vessel disease, are closely associated with cognitive outcomes, other studies suggest that mild degrees of small vessel disease (e.g., subcortical lacunes and white matter lesions) are not strongly associated with cognition; however, their presence may be associated with a lower threshold for cognitive impairment due to AD pathology [5], and hence, a higher risk for AD dementia [5,13]. Further, studies suggest that WMH are common in AD, and when present in parietal brain regions, are frequently associated with AD pathology rather than arteriolosclerosis risk [60]. Consistent with these findings, a recent study from the Harvard Aging Brain cohort showed that vascular risk and amyloid pathology interacted to predict more severe longitudinal atrophy in certain brain regions in AD; however, this interaction was not related to WMH. In individuals with a high amyloid burden, gray matter atrophy mediated the associations of vascular risk with cognitive outcomes [61].

Together, these findings suggest that vascular risk and Aβ likely have independent, yet overlapping [62], effects on brain atrophy and cognition [63] and that these effects are not strongly influenced by imaging estimates of small vessel disease [61]. Conversely, microvascular pathology in the form of capillary, endothelial, and pericyte abnormalities is ubiquitous in AD brains and may occur independently of vascular pathology in larger vessels [10,11]. There is growing evidence to support an important direct association of microvascular alterations with cognitive impairment and the onset or progression of AD pathology [8].

### 4.1. Brain Capillary Degeneration

Structural distortions to brain capillaries, including increased tortuosity, looping or kinking, basement membrane thickening, and luminal buckling or narrowing, are consistently observed in AD brains [10,16,17,18] and may have been reported in AD brains by Tuke decades prior to the first description of amyloid plaques and tau tangles [8,64]. It has been proposed that microvascular alterations in AD contribute to neuronal loss via reduced cerebral blood flow and impaired tissue oxygenation [8]. Microvascular alterations in AD display regional and laminar patterns that parallel those of neuronal loss [65], are preferentially localized in the vicinity of dystrophic neurites [66,67], and occur with a high prevalence in the hippocampus [65,68]. Animal studies have shown that significant alterations to brain microvasculature are evident in young AD transgenic mice and precede vascular or brain parenchymal amyloid pathology [15,69,70]. Furthermore, capillary abnormalities in AD do not correlate with the severity of AD pathological changes (i.e., Braak staging) [71,72], which suggests that microvascular alterations in AD are not solely a consequence of other AD pathologies.

### 4.2. Endothelium

Endothelial dysfunction is observed in almost all AD brains at autopsy [12], including the majority of AD brains that display no pathological evidence of macrovascular disease (i.e., arteriolosclerosis or atherosclerosis) [10,11]. Endothelial dysfunction refers to characteristic structural or functional alterations that occur at the intracellular or cell membrane level and interfere with endothelial cell integrity or function [9,10,11,16,65,66,67,73,74]. Structural changes consistent with endothelial dysfunction, such as increased pinocytic vesicles, laminin deposition in the basement membrane, collagen accumulation, and focal necrotic changes, are common in AD [9,11,16,73,74]. Endothelial cell loss in AD is demonstrated by reduced staining for endothelial markers, such as CD31 and CD34 [10], and higher plasma levels of endothelial markers, such as vascular cell adhesion molecule (VCAM-1), reflective of endothelial injury [75]. A recent study in a large well-characterized cohort of cognitively normal individuals with biomarker evidence of AD (i.e., “preclinical” or “pre-symptomatic” AD) has shown that cerebrospinal fluid (CSF) levels of the endothelial protein, vascular endothelial-cadherin (VE-Cadherin), are increased in even the earliest pre-symptomatic stages of AD compared to healthy controls; correlate with markers of amyloid, tau, and neurodegeneration; and are associated with cognitive outcomes independently of imaging measures of small vessel disease [21]. Furthermore, findings from this study suggest that endothelial injury partially mediates effects of amyloid and tau on cognitive outcomes [21].

Endothelial alterations have previously been considered secondary to amyloid or tau toxicity on cerebral vasculature; however, emerging evidence from animal studies suggests that endothelial dysfunction is an early and primary process in AD that may precede amyloid and tau pathologies and promotes abnormal protein aggregation and synaptic loss [12,15,69]. RNA-sequencing (RNA-seq) analyses suggest that endothelial pathways are among the most differentially expressed in human AD brains [76]. Consistent with these reports, 30 out of 45 AD risk genes are expressed in vascular cell types, and several of these have their highest expression levels in endothelial structures [77] (Table 1). Animal studies, including those that have examined the three-dimensional (3D) arrangement of brain vessels, suggest that significant morphological changes in cerebral vessels and endothelial dysfunction precede amyloid deposition and cognitive deficits in AD transgenic [69] or insulin-resistant mice [78]. While the mechanisms that underly these changes are yet to be elucidated, studies suggest that tumor necrosis factor (TNF)-α-mediated inflammation, oxidative stress, and leukocyte recruitment cause endothelial injury and loss of BBB integrity in insulin-resistant and diabetic mice [78,79]. It has been proposed that, in AD transgenic mice, impaired endothelial clearance of neuronal Aβ triggers the accumulation of Aβ and other proteins near endothelial membranes, which then form endothelial pores, further exacerbating endothelial dysfunction. Early endothelial damage correlates with cognitive deficits in these models and precedes the formation of larger vascular or parenchymal amyloid plaques [69]. Blocking endothelial interactions with leukocytes reduces Aβ deposition and tau hyperphosphorylation and improves memory in AD mouse models [80].

### 4.3. Pericytes

Pericytes are specialized cells located in the capillary basement membrane, which have contractile properties, and are uniquely positioned to interact closely with other cellular components of the neurovascular unit, such as endothelium, neurons, and astrocytes [81,82]. Pericytes play an important role in maintaining microvascular stability of the BBB and regulating neurovascular coupling and CBF [81]. Pericytes also demonstrate important phagocytic functions; under normal physiological conditions, these cells are involved in Aβ clearance and degradation through their surface expression of the low-density-lipoprotein receptor-related protein-1 (LRP-1) in an *APOE*-isoform-specific manner [83]; however, the phagocytic capacity of pericytes is limited in the presence of toxic Aβ levels, eventually leading to pericyte loss and further exacerbation of amyloid aggregation [82]. Pericyte loss is observed in AD and CAA mouse models and correlates with Aβ burden, white matter degeneration, and BBB disruption in the cortex and hippocampus [84,85,86,87].

Pericyte dysfunction has been associated with *APOE4*-mediated disruption of the neurovascular unit in clinical and animal studies. *APOE4* expression results in upregulation of the LRP-1-dependent proinflammatory cyclophilin A (CypA)–nuclear factor κB (NF-κB)– MMP-9 pathway in pericytes and brain endothelium. Activated MMP-9 from pericytes leads to endothelial toxicity, BBB breakdown, and neuronal injury [88]. An age-dependent disruption of the BBB is observed in *APOE4*-carriers years prior to the onset of cognitive decline and is associated with increased CSF levels of MMP-9 derived from pericytes [89].

Pericytes also appear to contribute to tau aggregation in AD; pericyte loss is associated with neuronal accumulation of hyperphosphorylated tau, caspase-cleaved tau, and tau aggregation in animal studies [90,91]. Interestingly, these changes are only observed in animal models that also had evidence of excess Aβ [91]. In a study by Sagare et al. [91], pericyte-deficient (*Pdgfrβ^+/−^*) mice were crossed with mice overexpressing the Swedish mutation of the human amyloid precursor protein (APP; *APP^sw/0^* mice) [91]. While neither *Pdgfrβ^+/−^* nor *APP^sw/0^* mice demonstrated tau pathology, *APP^sw/0^*/Pdgfrβ^+/−^ mice had evidence of tau pathology as early as 9 months. Neuronal loss and impaired hippocampal functions were observed in pericyte-deficient mice, although these changes appeared to be more severe when excess Aβ was also present [90,91]. Other animal studies have shown tau accumulation in endothelium and vascular smooth muscle cells in AD [92]. While impaired caveolin-mediated endocytosis in degenerating pericytes and vascular smooth muscle has been implicated in vascular tau accumulation in animal models of traumatic brain injury (TBI), it remains unclear whether similar mechanisms also occur in AD [93].

### 4.4. Astrocytic End-Foot Processes

Astrocytes are specialized glia that interact with other components of the neurovascular unit through long cellular processes (i.e., astrocytic end-feet) [94]. Several lines of evidence from in vitro and in vivo animal studies implicate astrocytes early in AD pathogenesis via several mechanisms [94], including increased Aβ synthesis and release, neuronal and astrocytic tau aggregation, neuroinflammation, increased oxidative stress, glutamate excitotoxicity due to disrupted astrocyte–neuron signaling, and impaired energy metabolism, which ultimately lead to neuronal and synaptic dysfunction.

Reactive astrocytes in AD may contribute to amyloid pathology through both *APOE4*-dependent and *APOE4*-independent pathways [95]. Under physiological conditions, astrocytes can internalize and degrade Aβ through enzymatic cleavage by neprilysin, insulin-degrading enzyme, and MMPs [95,96]. Astrocytes also contribute to Aβ clearance through the glymphatic system [97]. Astrocytic end-foot processes are rich in aquaporin-4 (AQP4), a transmembrane protein involved in regulating water transport and interstitial fluid/CSF exchange, including Aβ clearance. Loss of astrocytic AQP4 in AD is associated with impaired Aβ clearance [98].

Conversely, inflammatory conditions with high TNF-α and interferon-γ (INF-γ) levels are associated with increased Aβ synthesis and release by astrocytes [99,100]. Excess Aβ is toxic to astrocytes and activates several proinflammatory pathways, such as the receptor for advanced glycation end-products (RAGE) and NF-κB, which result in the release of inflammatory mediators into the neuronal milieu and promote Aβ synthesis [94]. *APOE4* expression by astrocytes also stimulates their transformation into the pro-inflammatory phenotype, which further contributes to Aβ aggregation [99].

Astrocytes have been linked to increased tau phosphorylation and tau-mediated neurodegeneration in AD, including tau accumulation within astrocytes and the release of tau oligomers through exosome-dependent pathways [101]. Astrocytic release of glypican-4 is implicated in *APOE4*-dependent tau hyperphosphorylation [102]. Several AD risk genes are expressed in astrocytes (*APOE4*, *WWOX*, *CLU*, *CDK2AP1*, *MEF2C*, and *IQCK*) and are linked to both amyloid and tau metabolism [95,103].

### 4.5. Other Vascular Constituents

Other vascular constituents, such as the extracellular matrix (ECM) and basement membrane proteins, may have a role in AD pathogenesis [104]. It has recently been shown that the ECM protein decorin is significantly increased in *APP* knock-in mice at an early age and that CSF decorin levels correlate with Aβ plaque load and decorin levels in the choroid plexus [104]. Consistent with these reports, CSF decorin levels are increased in preclinical AD and correlate with CSF Aβ42 levels [104]. Elevated CSF decorin levels appear to differentiate a subtype of AD characterized by activation of the innate immune system and possibly choroid plexus dysfunction [104].

## 5. Impaired Cerebral Perfusion Is a Common and Early Event in AD Which Strongly Predicts Future Cognitive Impairment in Clinical and Animal Studies

Impaired CBF, measured by arterial spin labeling magnetic resonance imaging (MRI), is well documented early in AD, including preclinical stages, as well as in animal studies [105,106,107,108,109]. Furthermore, CBF abnormalities predict future cognitive impairment and disease progression in symptomatic individuals [110]. CBF reductions of 10% to 25% are observed early in AD [111]. Studies utilizing single photon emission computed tomography (SPECT)-image reconstruction found that older adults with symptomatic AD have significantly reduced hippocampal perfusion compared to age-matched controls [112]. In one study, individuals with subjective memory loss who had significant hypoperfusion in the hippocampus and amygdala on SPECT were more likely to be diagnosed with AD over follow-up [113]. In another study of non-demented individuals, higher CBF was associated with larger hippocampal and amygdala volumes and lower risk for progression to dementia [114]. Consistent with these findings, lower volumetric flow rates in the internal and middle cerebral arteries, measured using four-dimensional (4D) MRI, were associated with more severe brain atrophy, and low flow rates in the internal carotid artery with amyloid pathology [115]. Temporoparietal, hippocampal–parahippocampal, posterior cingulate, and inferior parietal lobe hypoperfusion is more common in MCI or early symptomatic AD compared to controls, and several of these regional hypoperfusion abnormalities are associated with a higher risk for progression from MCI to AD dementia [112,116,117]. Importantly, recent data from large longitudinal AD cohorts enrolled in the Alzheimer’s Disease Neuroimaging Initiative (ADNI) suggests that vascular flow abnormalities occur very early in AD and may precede biomarker indicators of amyloid and tau pathology [19].

Consistent with these findings, animal models of chronic brain hypoperfusion (e.g., bilateral carotid artery occlusion) demonstrate AD pathology, including amyloid and tau aggregation and capillary changes in the CA1 region of the hippocampus that are almost identical to those seen in human AD brains [72]. In these models, CBF reductions to 25–33% of baseline were associated with several metabolic changes leading to cognitive and behavioral impairment and neuronal loss in the absence of strokes, hemorrhages, or hypertension [118,119]. These changes included loss of microtubule-associated protein-2 in CA1 [120], reduced hippocampal cytochrome oxidase [121], impaired monoamine neurotransmitter turnover [108], altered glucose utilization [122], reduced post-synaptic cholinergic activity [123], increased heme oxygenase [124] and matrix metalloproteinase (MMP)-2 expression [125], and reactive gliosis, leading to a state of impaired energy metabolism, disturbed protein synthesis, and increased oxidative stress, which were detectable months prior to the onset of hippocampal neuronal loss or cognitive impairment. Other animal studies of chronic cerebral hypoperfusion show increased hippocampal and frontal Aβ pathology due to activation of β- and γ-secretases [126,127] (two enzymes in the amyloidogenic pathway of *APP* processing that sequentially cleave *APP* to produce the Aβ42 peptide), and impaired Aβ clearance due to reduced LRP-1 and increased RAGE expression in the brain endothelium [128].

While neuronal dysfunction and subsequent hypometabolism may contribute to reduced CBF in AD, several observations suggest that hypoperfusion in AD cannot be fully attributed to hypometabolism [129]; CBF alterations are detected before neurodegeneration [19] and the degree of hypoperfusion in AD exceeds what is expected based on the degree of atrophy alone [130]. Importantly, structural and functional changes to brain capillaries which are observed in AD, including capillary constrictions near amyloid deposits, and contraction of pericytes and endothelium, have been shown to reduce blood vessel diameter by up to 50% and thereby provide a mechanistic basis for reduced CBF in AD beyond what can be explained by neuronal or blood vessel loss [131,132]. Other possible mechanisms for capillary constriction in AD include regional vasoconstriction and increased vascular resistance in response to Aβ [133], capillary blockage by neutrophils [134], fibrin deposits [135], and focal constriction sites caused by pericyte soma [133]. Administration of an antineutrophil antibody (anti-Ly6G), or treatment with the anticoagulant dabigatran, are associated with increased CBF and improved memory functions in transgenic AD models [134,135]. High levels of reactive oxygen species (ROS) produced by microglia and pericytes, and endothelin-1 produced by the endothelium, are associated with capillary constriction in vitro [133]. Endothelial dysfunction may contribute to reduced CBF due to imbalance between vasodilators (e.g., nitric oxide (NO), bradykinin, and prostacyclin) and vasoconstrictors (e.g., endothelin-1 and thromboxane A2) [14]. Oligomeric Aβ further exacerbates capillary constriction and reduced CBF via the activation of microglial and astrocytic inflammasomes and the release of interleukin (IL)-1β. Furthermore, low levels of neuroglobin, a protein with ROS-scavenging and anti-apoptotic activities, may contribute to increased sensitivity of the hippocampus to hypoxia and, subsequently, an increased regional vulnerability to AD pathology due to impaired oxidative metabolism [136]. Together, these various mechanisms lead to hypoxia and reduce CBF, which contributes to neuronal dysfunction, synaptic loss, and hypometabolism [129,130].

Additional support for the importance of vascular contributions to AD pathogenesis comes from studies of individuals with trisomy 21 (Down syndrome) who have an additional copy of the *APP* gene. Capillary degeneration [65] and reduced brain perfusion are observed in the brains of individuals with Down syndrome years prior to any evidence of amyloid plaques or neurofibrillary tangles [137,138]. Oxidative stress appears to precede Aβ deposition by many years in individuals with Down syndrome who die in their teens and twenties [139], a finding that indicates that AD-like pathology is not likely the trigger of neuronal metabolic disruption in these individuals.

## 6. A Forward-Feedback Loop between Hypoxia and Endothelial Dysfunction Contributes to AD Pathogenesis

The brain endothelium is a highly active interface that regulates the neuronal milieu via diffusible mediators and, as a part of the neurovascular unit, controls blood flow and neuronal metabolism [14,140]. Brain endothelial cells produce several extracellular matrix proteins (e.g., proteoglycans, laminin, and fibronectin), growth factors (e.g., endothelium-derived growth factor (EDGF), platelet-derived growth factor (PDGF), and fibroblast growth factors (FGF)), proteases (e.g., thrombin and MMPs), vasoconstrictors (e.g., endothelin-1, leukotrienes, and thromboxane A2) and vasodilators (e.g., NO, prostacyclin/prostaglandin E2, and endothelium-derived hyperpolarizing factor (EDHF)), antithrombotic (e.g., antithrombin III) and prothrombotic factors (e.g., thromboplastin and platelet-activating factor), and various inflammatory mediators (e.g., IL-1, IL-6, IL-8, monocyte chemoattractant proteins (MCP) 1-2, leukotrienes, and cell adhesion molecules) [14,140].

Several studies suggest that hypoxia causes structural and functional alterations in the brain endothelium via different mechanisms, including rearrangement of the endothelial cytoskeleton, altering intercellular adhesions of endothelial cells, and promoting endothelial constriction, apoptosis, and degeneration [141]. Studies of intermittent hypoxia suggest that endothelial cell resistance is reduced due to ROS-dependent activation of the extracellular signal-regulated kinase (ERK1/2) pathway and c-Jun N-terminal kinase (JNK)-mediated re-arrangement of the endothelial cytoskeleton, including the redistribution of actin stress fibers and VE-cadherin, leading to abnormal endothelial barrier function [142,143]. Hypoxia stimulates the release of inflammatory mediators (e.g., thrombin, histamine, and bradykinin), which contribute to increased endothelial cell contractibility and permeability [144]. In vitro studies suggest that hypoxia increases endothelial apoptosis and endothelial cell degeneration by stimulating their transition into fibroblast-like cells [145]. Hypoxia is also associated with activation of the pro-inflammatory Toll-like receptor-4 (TLR4)–NF-κB pathway [146] and interferes with the physiologic anti-inflammatory and anti-oxidant properties of the endothelial membrane constituent, high-molecular-weight hyaluronic acid [147]. Conversely, endothelial dysfunction further exacerbates CBF reductions and cellular hypoxia through several mechanisms such as impaired angiogenesis, dysregulated thrombosis, local immune dysregulation, vascular inflammation, and neurovascular uncoupling [14]. Together, these findings suggest that hypoxia-ischemia and endothelial dysfunction act in a forward-feedback loop to trigger and exacerbate AD pathology, including abnormal protein aggregation, vascular inflammation, and neuronal/synaptic injury, via several converging mechanisms that are described below (Figure 1).

### 6.1. Increased Amyloidogenic APP Processing

Data from animal and human studies suggests that capillary constriction in AD is sufficient to reduce CBF by up to 50%, leading to upregulation of the β-secretase1 (*BACE1*) enzyme involved in Aβ synthesis [107]. *BACE1* levels are increased in human AD brains, including those in the preclinical or early symptomatic stages (i.e., MCI), and may correlate with Aβ plaque burden and cognitive decline [148]. Mechanisms by which reduced CBF increases *BACE1* expression in AD include a hypoxia-inducible factor (HIF)-1α-mediated increase in *BACE1* mRNA expression and enzymatic activity [149]. The *BACE1* gene promoter contains a functional hypoxia-response element (HRE), and HIF-1α can effectively bind to the *BACE1* promoter [149]. Significant reductions in cortical and hippocampal *BACE1* levels are observed in HIF-1α knockout mice. *BACE1* upregulation due to hypoxia is also mediated by caspase-3, which cleaves GGA3 (Golgi-associated, gamma adaptin ear containing, ARF binding protein-3) involved in *BACE1* trafficking, thereby reducing *BACE1* degradation [150,151,152]. As a γ-secretase-activating protein, HIF-1α also interacts directly with γ-secretase and increases its activity [153]. The APP intracellular domain (AICD) peptide, a product of *APP* cleavage by γ-secretase, stimulates the transcription of the *HIF-1*α gene, potentially creating a positive forward loop between HIF-1α and γ-secretase [154]. Hypoxia alters several proteins that interact with γ-secretase, including voltage-dependent anion channel 1 (VDAC1), a γ-secretase-activating protein whose expression is increased in human AD brains and AD mouse models [155]. Reduced VDAC1 expression is associated with lower brain expression of *APP*, *Presenilin-1* (*PS1*), *Presenilin-2* (*PS2*), and *BACE1* [156]. Other studies have shown that hypoxia inhibits α-secretase, an enzyme in the non-amyloidogenic pathway of *APP* processing, which reduces the production of the extracellular s*APP*α fragment, and results in subsequent loss of its neuroprotective and neurotrophic effects [149]. Studies in neuronal cultures suggest that hypoxia increases the production of an alternative splice variant of *PSEN2* (aka *PS2*) that lacks exon 5, referred to as *PSEN2V* [157], via activation of the high mobility group A protein 1a (HMGA1a). The *PSEN2V* variant is abundantly expressed in the hippocampus of AD brains, promotes Aβ synthesis, and increases the vulnerability of the endoplasmic reticulum (ER) to stress [157].

### 6.2. Activation of Tau Kinases

Tau hyperphosphorylation is increased under hypoxic-ischemic conditions in an Aβ-independent manner [158,159]. Tau hyperphosphorylation is observed in hypertensive rats that do not display Aβ pathology [158] and can be triggered by unilateral carotid artery occlusion even when Aβ levels are not elevated [159]. Proposed mechanisms include activation of Cdk5 (cyclin-dependent kinase-5), calpain-mediated cleavage of the Cdk5 regulatory subunit p35 [160,161], and activation of glycogen synthase kinase (GSK)-3β via reduced activity of the phosphatidylinositol 3-kinase/Akt pathway [162].

### 6.3. Endoplasmic Reticulum Stress

ER stress and impaired autophagy also occur due to hypoxia and are associated with increased *PSEN1* (aka *PS1*) expression and γ-secretase activity, contributing to increased amyloid pathology [149], and m-calpain activation which promotes tau phosphorylation [163]. Two major pathways which are involved in the ER-stress-induced unfolded protein response (UPR), PERK–eIF2α–ATF4 and IRE1–X-box binding protein 1 (XBP1) [164], are activated in the setting of hypoxia-ischemia. ATF4 binds to the *PSEN1* gene promoter and induces its transcription [165], and PERK-eIF2α signaling increases *BACE1* expression [166]. ER stress can also suppress autophagy through different mechanisms (e.g., increased degradation of the autophagy inducer FOXO1 by XBP1), resulting in increased Aβ synthesis [149,167].

Endothelial dysfunction also contributes to increased amyloid and tau aggregation via different mechanisms, including ineffective Aβ and tau clearance, dysregulated lipid raft endocytosis, and reduced endothelial nitric oxide synthase (eNOS), which are described below.

### 6.4. Impaired Endothelial Aβ Clearance

Two neuronal cell surface receptors involved in Aβ clearance are impaired in AD: the downregulation of LRP-1, which transports free parenchymal Aβ into the systemic circulation, and the upregulation of RAGE, which transports Aβ across the BBB into the brain [168]. Several reports from animal and neuropathological studies suggest that vascular LRP-1, expressed by endothelium, pericytes, and vascular smooth muscle, is also downregulated in AD and contributes to impaired Aβ clearance [169]. Further, Aβ toxicity is associated with increased proteasomal degradation of LRP-1 in endothelium [170]. Inhibition or knock-down of vascular LRP-1 in animal models is associated with increased parenchymal and vascular Aβ deposits [171]. Aβ accumulation in AD is further exacerbated by reduced levels of the endothelial molecule, PICALM (phosphatidylinositol-binding clathrin assembly protein), which is important for LRP-1-mediated Aβ-transport [172]. Conversely, vascular RAGE expression is increased in AD, especially in areas with high Aβ pathology, and contributes to increased brain amyloid load. RAGE inhibition reduces Aβ pathology and improves cognition in AD mouse models [173]. Brain endothelial cells also express Aβ-degrading enzymes, such as neprilysin and insulin-degrading enzyme whose expression or activity is altered in AD, further promoting Aβ aggregation [174]. Proteoglycans on the endothelial cell surface interact directly with Aβ peptides via electrostatic interactions promoting Aβ fibrillization and plaque formation and promote the uptake of extracellular *APOE4*-containing lipoproteins [175].

### 6.5. Dysregulated Endothelial Nitric Oxide Synthase (eNOS)

Endothelial NOS (eNOS) plays an essential role in tau homeostasis and regulating tau phosphorylation by balancing the activity of tau kinases and phosphatases [176]. Crossing *eNOS* knockout (*eNOS*^−/−^) mice with a mouse model of amyloid pathology (*APP*/*PS1* mice) was associated with increased phosphorylated tau (p-tau), which was attributed to an increase in Cdk5 activation by p25, while other kinases, such as GSK-3β, were unchanged [176,177]. Interestingly, neuronal p25 levels and Cdk5 activity were increased in the *eNOS*^−/−^ and the *APP*/*PS1*/*eNOS*^−/−^ mice, supporting the notion that the brain endothelium can influence neuronal mechanisms via soluble mediators [176,177]. Another important finding reported by these studies was that *eNOS* knockout alone was not sufficient to induce tau pathology, as increased p-tau was only observed in *eNOS* knockout mice that also had amyloid pathology (i.e., *APP*/*PS1*/*eNOS*^−/−^ mice) [176,177]. This is consistent with the current AD model, which suggests that Aβ pathology precedes tau pathology [3] and is a prerequisite for AD pathogenesis [178]; however, tau-dependent processes are needed for Aβ-mediated synaptic toxicity [179]. These findings propose additional mechanisms by which endothelial dysfunction may contribute to AD pathology via increased tau phosphorylation and enhancing Aβ-mediated toxicity.

### 6.6. Altered Lipid Raft Endocytosis

Altered lipid raft endocytosis has been implicated in abnormal protein aggregation in AD and other neurodegenerative disorders and has been proposed as an additional mechanism by which hypoxia or endothelial dysfunction contribute to AD pathogenesis [180,181]. Lipid rafts are specialized microdomains of cell membranes rich in cholesterol, sphingolipids, and saturated fatty acids and play an important role in regulating membrane trafficking, ligand binding, axonal growth, and synaptic maintenance [182]. Profound changes in neuronal lipid raft composition and dynamics are detectable in even the earliest preclinical stages of AD [180]. Lipid raft endocytosis facilitates Aβ synthesis by β- and γ-secretases [183,184] and promotes amyloid aggregation [185]. Several lines of evidence implicate altered lipid raft endocytosis in abnormal Aβ accumulation in AD, including increased β- and γ-secretase *APP* processing within the lipid rafts [183,184], and increased translocation of active *PS1* and *NOTCH3* into lipid drafts due to oxidative stress [186]. Aβ in lipid rafts is associated with increased recruitment of *APOE4* and p-tau to the membrane in an age-dependent manner [187]. Interestingly, while tau is considered a neuronal protein, most of tau is secreted in a vesicle-free form which can interact with the cell membrane via LRP-1, heparan sulfate proteoglycans, and direct binding to membrane lipids [188], and p-tau filaments have also been shown to accumulate within lipid rafts [188].

Lipid raft clusters are also present in endothelial cells and contribute to vascular inflammation [189]. Further, vascular endothelial growth factor receptor-2 (VEGFR2), an angiogenic receptor that is expressed on endothelial cells, colocalizes with endothelial lipid rafts to regulate its activation [181]. Lipid rafts are important in modulating the stability of non-activated VEGFR2, thereby stabilizing vascular endothelial growth factor (VEGF)-mediated activation of the ERK pathway, which is important for angiogenesis [181]. Oxidative stress activates apoptotic pathways and increases lipid raft clustering in endothelial cells which stimulates NAD(P)H-oxidase-generated ROS and leads to endothelial dysfunction [190].

### 6.7. Impaired Angiogenesis

Angiogenesis and vascular remodeling are impaired in AD as evidenced by altered expression profiles of several angiogenic factors, such as VEGF, vascular-restricted mesenchyme homeobox-2 (MEOX2), MMP-9, angiopoietin-2, integrins (αvβ3 and αvβ5), and the transferrin receptor, leading to vascular regression and abnormal vessel sprouting [191,192]. Several inflammatory mediators (e.g., IL-1β) stimulate angiogenesis through pathways that overlap with hypoxia signaling, suggesting important links between hypoxia, inflammation, and angiogenesis in AD [14].

Although several angiogenic factors are upregulated in AD, Aβ toxicity and low MEOX-2 levels in the brain endothelium, which impair angiogenesis, are also present. Hypoxia suppresses MEOX-2 expression in brain endothelium, and MEOX-2-deficient mice demonstrate vascular regression [193]. Therefore, new vessel formation in AD is ineffective and dysregulated, causing aberrant vessel formation, premature vessel pruning, and reduced microvascular density [194,195]. Hypoxia may promote aberrant angiogenesis by upregulation of HIF-1α expression, and impaired angiogenesis further exacerbates reduced CBF and its subsequent effects on amyloid and tau aggregation. Endothelial dysfunction is implicated in dysregulated angiogenesis due to loss of regulatory feedback signals, leading to a state of chronic endothelial activation, which further contributes to hypoxia and neuronal damage [14].

VEGF is an important pro-angiogenic factor that is present in the vessel wall, reactive astrocytes, and perivascular amyloid deposits [196]. While a few animal studies suggest that VEGF is downregulated in AD transgenic mice [197], most animal and clinical studies demonstrate that VEGF brain expression levels are increased in AD and are associated with cognitive decline [198]. Elevated intrathecal VEGF levels correlate with amyloid pathology and worse cognitive outcomes [199]. Several polymorphisms in the VEGF gene promoter have been associated with a higher AD risk [200]. Despite increased VEGF expression, studies suggest that it is ineffective in promoting angiogenesis in AD [201], as Aβ toxicity interferes with VEGF binding to its receptor, VEGFR-2, on endothelial cells, thereby interfering with its pro-angiogenic role. Paradoxically, increased VEGF expression in AD appears to exacerbate CBF reductions by causing capillary stalls and reductions in the tight junction protein occludin, leading to increased vascular permeability, the subsequent activation of endothelial inflammatory pathways, and the recruitment of leukocytes towards injured endothelium, which exacerbates capillary blockage and the stalling of blood flow [202].

### 6.8. Dysregulated Thrombosis

Thrombin is an endothelium-derived factor with important roles in angiogenesis and inflammation and contributes to tau pathology and neurodegeneration in AD. High levels of thrombin and the thrombin receptor, protease-activated receptor-1 (PAR-1) [203,204], and low levels of the thrombin inhibitor protease nexin-1 [205], are observed in AD brains. High thrombin levels in AD are associated with the upregulation of several endothelial proteins including VEGF receptors, angiopoietin-2, αvβ3, MMPs, IL-1β, and IL-8 [206,207,208]. Thrombin is directly toxic to neurons in vivo and in vitro [209,210]. Intracerebral injection of thrombin or overexpression of the thrombin receptor PAR-1 are associated with neurotoxicity and cognitive deficits in animal studies [211] via several mechanisms, including increased tau aggregation, apoptosis, oxidative stress, and microglial or astrocytic activation [212,213,214,215]. Thrombin activates pro-inflammatory pathways in microglia and astrocytes; these include the JAK2–STAT3 pathways, which increase the production of TNF-α and inducible NOS in microglia, and regulation of ERK1/2 leading to increased expression of MMP-9 in astrocytes [215,216,217].

### 6.9. Vascular Inflammation and Immune Dysregulation

Recent studies have shown that endothelial cells and pericytes demonstrate immunoregulatory functions and influence the neuronal milieu through the release of inflammatory and immune mediators. The brain endothelium in AD expresses high levels of inflammatory mediators such as MCP-1, intercellular adhesion molecule-1 (ICAM-1; CD54), and cationic antimicrobial protein 37 kDa (CAP37) [14,218,219]. Microvessels from AD brains secrete significantly higher levels of IL-1β, IL-6, IL-8, MMPs, thrombin, TNF-α, and transforming growth factor (TGF)-β compared to microvessels from healthy brains [14,218,219]. Aβ upregulates C-C chemokine receptor type 5 (CCR5) expression by the brain endothelium via its interaction with RAGE receptors and the activation of JNK-, ERK-, and PI3K-signaling pathways [220]. E-selectin (CD62) is an inducible endothelium-specific protein that is exclusively expressed in response to endothelial cell activation by cytokines [221,222]. Stimulation of endothelial Toll-like receptor 2 (TLR2) is associated with the upregulation of E-selectin and the recruitment of immune cells [223]. A few studies have shown that CSF E-selectin levels are elevated in dementia and correlate with AD biomarkers [224]. Furthermore, inhibition of vascular inflammation by sunitinib has been shown to reduce vascular Aβ pathology and improve cognitive deficits in AD animal models [225]. Endothelial cells may also act as antigen-presenting cells and express major histocompatibility complex (MHC) proteins I and II [226], and a subset of endothelial cells has been shown to perform immune functions, such as phagocytosis and scavenging, when stimulated by Aβ, including surface expression of the scavenger receptor CD36 [227,228].

### 6.10. Neuronal and Synaptic Dysfunction

Endothelial dysfunction indirectly contributes to neuronal and synaptic loss via increased protein aggregation, vascular inflammation, impaired angiogenesis, and dysregulated thrombosis; however, it has become increasingly recognized that endothelial injury can also cause neuronal and synaptic loss via other mechanisms, including loss of endothelial GLUT-1 (glucose transporter protein type 1) which leads to impaired neuronal glucose uptake and metabolism, impaired neurovascular coupling, and potential direct endothelial interactions with pathways involved in synaptic plasticity and axonal growth and repair [21,229].

There is growing evidence from animal and translational AD studies that endothelial dysfunction exerts direct effects on synaptic and neuronal functions that are independent of amyloid or tau [14]. For example, endothelial dysfunction is associated with reduced expression of several presynaptic (e.g., synaptosomal-associated protein-25 (SNAP-25) and growth associated protein-43 (GAP-43)) and postsynaptic (e.g., PSD95) proteins in human AD brains [230,231,232]. Proposed mechanisms by which endothelial dysfunction predisposes to neuronal/synaptic loss include impaired release of neurotrophic growth factors (e.g., VEGF and FGF), and reduced hippocampal expression of NO, an important mediator of synaptic transmission and long-term potentiation [12,14].

Recent studies highlight several pathways by which endothelial dysfunction influences axonal repair, neuronal integrity, and synaptic plasticity independently of amyloid and tau, including interactions of endothelial proteins with semaphorin-3A involved in memory or learning functions [21,22]. Endothelial proteins also interact with the neuropilin/plexin-A1 (NRP-1/PLXNA) complex and Nr-CAM (Figure 2), which are expressed on neuronal surfaces and regulate axonal growth during development or in response to injury [233]. Interestingly, recent studies in animal stroke models suggest that endothelial cells regulate astrocytic differentiation into neural progenitor cells and improve behavioral recovery through the upregulation of the pro-neural factor Ascl1 by the endothelium [234].

While the mechanisms that underly neurovascular uncoupling in AD are complex, there is ample evidence to suggest that brain endothelial dysfunction plays an important role in this process [235,236]. NADPH oxidase and the generation of ROS impair endothelial NO production and cause an imbalance between endothelial vasodilator and vasoconstrictor responses [237]. Further, insulin-like growth factor (IGF-1) deficiency has been shown to promote endothelial and astrocytic dysfunction in AD, including dysregulation of metabotropic-glutamate-receptor expression in astrocytes and impaired CBF responses mediated by eicosanoids [238]. Neurovascular uncoupling due to IGF-1 deficiency in animal studies is associated with deficits in hippocampal-dependent spatial memory, suggesting an important association between neurovascular coupling and cognition [238].

**Figure 2 biomolecules-13-00830-f002:**
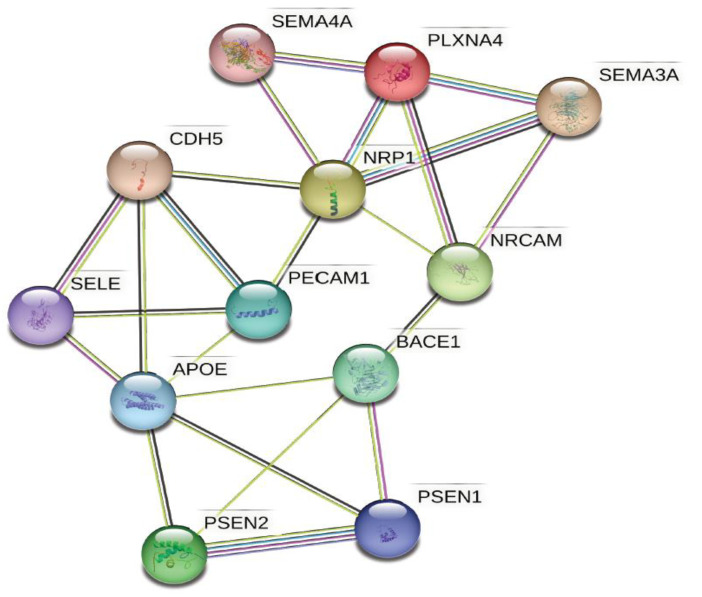
**Endothelial proteins interact with molecular pathways involved in synaptic plasticity and axonal repair.** Functional pathway analyses of endothelial and neuronal proteins in AD using STRING. These functional pathway analyses suggest significant interactions between endothelial and neuronal proteins. Network nodes represent proteins; splice isoforms or post-translational modifications are collapsed so that each node represents all the proteins produced by a single protein-coding gene locus. Edges represent protein–protein associations that are meant to be specific and meaningful. Color coding for the interactions are as follows: known interactions from curated databases (teal); known experimentally determined interactions (purple); predicted interactions from gene neighborhood (green), gene fusions (red), or gene co-occurrence (blue); predicted interactions from text-mining (yellow), co-expression (black), and protein homology (lavender). *APOE*, apolipoprotein E; *BACE*, β-secretase; *CDH5*, cadherin-5, also known as vascular endothelial-cadherin (VE-cadherin); *NRCAM*, neuronal cell adhesion molecule; *NRP-1*, neuropilin-1; *PECAM1,* platelet endothelial cell adhesion molecule-1, also known as CD31; *PSEN1*, presenilin-1 (aka PS1); *PSEN2*, presenilin-2 (aka PS2); *PLXNA4*, Plexin A4; *SELE*, E-Selectin; *SEMA3A*, semaphorin-3A; *SEMA4A*, semaphorin-4A. Figure created with STRINGv11.5 for functional protein association networks (https://string-db.org) [239,240,241].

### 6.11. Impaired Neurogenesis

In addition to neurons, brain endothelial cells also produce brain-derived neurotrophic factor (BDNF), which plays an important role in neurogenesis and neuroplasticity [242]. One mechanism by which endothelial dysfunction can contribute to neurodegeneration is through impaired neurogenesis and loss of the neurotrophic and neuroprotective effects of endothelial BDNF. Animal models of DM and hypertension demonstrate reduced neuronal and endothelial BDNF expression, which is considered a marker of endothelial dysfunction [242,243,244]. Loss of endothelial-derived BDNF decreases neuronal resistance to conditions associated with oxidative stress such as hypoxia, glucose deprivation, and Aβ toxicity [245,246].

Proposed mechanisms by which endothelial BDNF promotes neuroplasticity include direct binding to neuronal tyrosine receptor kinase B (TrKB) receptors or sustained activation of endothelial TrKB receptors, which results in increased NO production and induces long-term potentiation, an essential process for memory and learning [247,248]. The latter suggests a direct link between endothelial dysfunction and cognitive impairment. Further, some of the effects of endothelial BDNF on neurons may be mediated by astrocytes at tripartite synapses. Astrocytes express the p75 neurotrophin receptor (p75NTR), which binds with high affinity to the BDNF precursor, proBDNF, leading to the internalization of this complex into astrocytes and the maturation of proBDNF into mature BDNF, which is then released into the neuronal milieu [249,250]. Therefore, endothelial BDNF can modulate synaptic plasticity through astrocyte-dependent or astrocyte-independent pathways (Figure 3).

Together, these findings support the role of dysfunctional endothelium in mediating neurodegeneration in AD. Further support for this comes from studies that show that the direct co-culture of neurons with microvessels from AD brains, or the exposure of cultured neurons to conditioned mediums from AD microvessels but not microvessels from normal aging brains, is associated with neurotoxicity [251], and that brain endothelial mechanisms (e.g., endothelial BDNF) are directly associated with neuroplasticity and cognition. Importantly, data from animal and clinical studies suggests that endothelial and vascular alterations, leading to reduced CBF and capillary loss, may be the inciting event in AD pathogenesis, which then triggers the AD cascade (Figure 1). This posits the notion that AD may primarily be an endotheliopathy; however, additional mechanistic studies in animal models and cell cultures will be needed to support this hypothesis. Further, longitudinal clinical studies that measure fluid and imaging biomarkers of various AD pathologies (including endothelial injury) in well-characterized AD cohorts will provide important insight into the significance of endothelial dysfunction and the temporal ordering of endothelial dysfunction in relation to other AD pathologies and the onset of cognitive impairment.

Finally, improving brain endothelial health has the potential to support neuronal regeneration and repair. Endothelial progenitor cells (EPCs), which express the surface markers CD34, VEGFR2, and CD133, are bone-marrow derived cells that can differentiate into mature endothelial cells and produce angiogenic growth factors to support endothelial repair [252]. Reduced EPC counts and functionality (e.g., reduced mobility or adhesion capacity) have both been reported in AD and correlate with the severity of cognitive impairment [253,254]. EPCs are targeted by Aβ toxicity, further exacerbating endothelial dysfunction by impairing endothelial repair [254]. Conversely, transplantation of EPCs is associated with upregulation of tight-junction proteins, increased angiogenesis, higher hippocampal neuronal survival due to anti-apoptotic effects, reduced hippocampal and cortical Aβ, and improved memory and learning functions [255]. Animal studies provide exciting preliminary evidence to support the utility of EPC administration in restoring the BBB and promoting angiogenesis in animal models of stroke and TBI.

## 7. Conclusions and Future Directions

Although the current hypothesis of neurodegeneration in AD is centered around abnormal amyloid and tau aggregation, emerging evidence suggests that endothelial dysfunction is an early and primary event in AD pathogenesis that may precede abnormal protein aggregation and directly contribute to neurodegeneration and synaptic injury. Together, studies examining endothelial contributions to neuronal and synaptic dysfunction in AD suggest it may be a potential therapeutic target in AD. Brain endothelial health has both direct and indirect effects on neuronal and axonal repair mechanisms and synaptic plasticity via endothelial–neuronal interactions as well as diffusible endothelial mediators that regulate neuronal functions. Therapeutic interventions, which improve endothelial health and promote endothelial repair, have the potential to improve neurovascular coupling, reduce vascular inflammation and thrombosis, and decrease amyloid and tau aggregation, ultimately leading to restoring CBF and improving cognition. Specifically, the implementation of EPCs may offer an exciting opportunity for endothelial repair and mitigating the deleterious effects of endothelial dysfunction early in the disease process. Further investigations into the mechanisms by which endothelial repair may improve neuronal and synaptic plasticity, and the development of novel AD therapies targeting endothelial injury in AD, are needed.

## Figures and Tables

**Figure 1 biomolecules-13-00830-f001:**
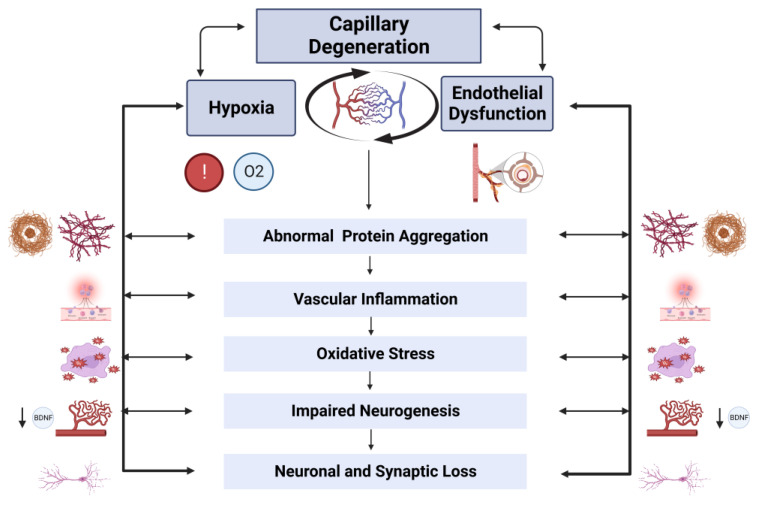
**Bidirectional interactions between hypoxia and endothelial dysfunction in Alzheimer Disease.** Capillary degeneration leads to reduced cerebral blood flow and hypoxia, which causes endothelial injury, further exacerbating tissue hypoxia. A forward-feedback loop between hypoxia and endothelial dysfunction initiates a cascade of events, including amyloid and tau aggregation, vascular inflammation, oxidative stress, impaired neurogenesis, and neuronal and/or synaptic injury. Created with Biorender.com.

**Figure 3 biomolecules-13-00830-f003:**
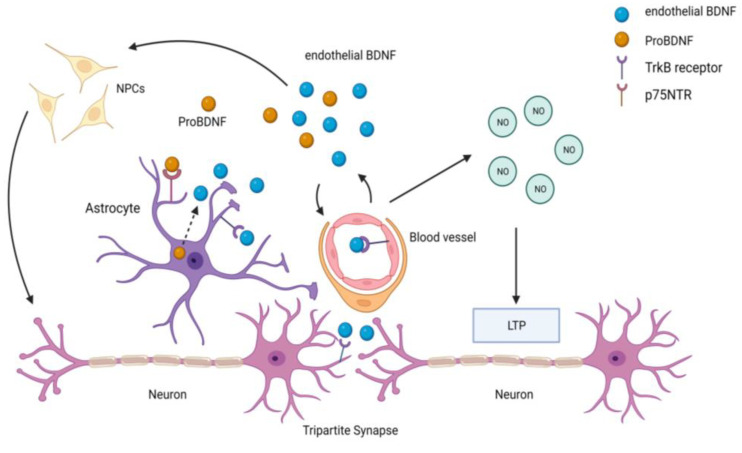
**Schematic diagram of the putative role of endothelial-derived BDNF in neurogenesis.** Endothelial cells produce BDNF, which binds to TrkB receptors on endothelium, neurons, and astrocytes. Endothelial-derived BDNF can influence neuronal functions and differentiation through astrocyte-dependent and -independent pathways. Endothelial cell activation by BDNF increases nitric oxide (NO) production, which is important for long-term potentiation (LTP) and memory functions. Endothelial-derived BDNF stimulates the differentiation of neural progenitor cells (NPCs) into neurons. Astrocytes at tripartite synapses can also internalize the BDNF precursor pro-BDNF, through the p75 neurotrophin receptor (p75 NTR), and transform it into mature BDNF before releasing it into the extracellular space. Created with Biorender.com.

**Table 1 biomolecules-13-00830-t001:** AD Risk Genes Expressed in Brain Vascular Cell Types.

AD Risk Gene	Vascular Cell Types
*APOE*	Smooth muscle cells, Meningeal fibroblasts
*PICALM **	Brain endothelium (arterial, capillary, and venous)
*CLU*	Meningeal fibroblasts, Ependymal cells
*ABCA7*	T cells
*PTK2B*	T cells
*PLCG2 **	Brain endothelium (arterial)
*HLA-DRB1 **	Brain endothelium (arterial)
*CD2AP **	Brain endothelium (arterial, capillary, and venous)
*SLC24A4*	Ependymal cells
*RIN3*	T cells
*ADAMTS1 **	Smooth muscle cells, Pericytes, Brain endothelium (arterial)
*ADAMTS4*	Smooth muscle cells, Pericytes
*FERMT2*	Smooth muscle cells, Pericytes
*SCIMP*	Ependymal cells
*CLNK*	T cell
*ECHDC3*	Perivascular fibroblasts
*TNIP1 **	Brain endothelium (capillary and venous), T cells
*ABCA1 **	Brain endothelium (venous), Perivascular fibroblasts
*USP6NL **	Brain endothelium (capillary and venous)
*INPP5D* *	Brain endothelium (capillary), T cells
*ACE **	Brain endothelium (arterial and capillary)
*IQCK*	Ependymal cells
*ABI3*	T cells
*HESX1*	Meningeal fibroblasts
*FHL2*	Perivascular fibroblasts
*CHRNE*	Perivascular fibroblasts, T cells
*AGRN*	Pericytes
*IL34*	Smooth muscle cells, Pericytes, Meningeal fibroblasts
*NYAP1*	Ependymal cells
*CASS4 **	Brain endothelium (capillary and venous)

Of the 45 top AD risk genes, 30 genes (shown above) are expressed in vascular cell types (excluding perivascular macrophages), including 11 that are expressed in the brain endothelium (marked with an asterisk). The remaining 15 genes include 6 genes that are solely expressed in perivascular macrophages (*CR1*, *HAVCR2*, *CD33*, *TREM2*, *MS4A6A*, and *SPI1*). The other 9 top AD risk genes were *BIN1*, *SORL1*, *ADAM10*, *CNTNAP2*, *WWOX*, *APH1B*, *HS3ST1*, *KAT8*, and *CCDC6*. Adapted from Yang et al. [77].

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
