# Peer review of "Microvascular Contributions to Alzheimer Disease Pathogenesis: Is Alzheimer Disease Primarily an Endotheliopathy?"

_biomolecules, 2023, doi:10.3390/biom13050830_

Round 1

Reviewer 1 Report

The manuscript by Dr. Rawan Tarawneh is a nicely written review on the role of vascular constituents and vascular complications in Alzheimer disease pathogenesis. 

However, I have a couple of suggestions/ questions: 

1.     Reference(s) required in lines 59-64. 

2.     Line 68, what was the hypothesis proposed 30 years ago and provide an original reference. 

3.     Line 92, introduce APOE4 and a brief description about how APOE4 is involved in AD pathology, and it is strongest risk factor for AD/dementia development. 

4.     Lines 96-103, it be beneficial to include studies and mechanism about how LDL, homocysteine are linked to AD. 

5.     Line 154-157, provide an original reference rather than a review paper. 

6.     Regarding lines 187-188, it be beneficial to provide the list of 45 AD risk and their site of expression. 

7.     Reference(s) required in line 230. How are astrocytes involved in Aβ synthesis? Is it via primarily via APOE4?

8.     Introduce β- and γ-secretase and their function in line 281. 

9.     Reference(s) required in lines 309-311. 

10.  Figure legend missing in both figures. 

11.  What is the role of VDAC1?

12.  Reference(s) required in lines 414-416, 429-430, and 436-438. 

13.  Information in section 5.6 is linked to neuronal lipid raft or endothelial.

14.   Introduce GLUT1

15.  Is Figure 2 taken from previous published manuscript or newly generated. Provide details in figure legend. 

Author Response

The manuscript by Dr. Rawan Tarawneh is a nicely written review on the role of vascular constituents and vascular complications in Alzheimer disease pathogenesis. 

However, I have a couple of suggestions/ questions: 

Response: I thank the reviewer for this positive feedback and thoughtful review of the manuscript.

I have addressed all reviewer comments here and in the revised document (marked in blue).

  1. Reference(s) required in lines 59-64. 

Response: References 8, 9,13-15,19, and 20 have been included for those lines 60-66.

  1. Line 68, what was the hypothesis proposed 30 years ago and provide an original reference. 

Response: This has now been clarified with original references 23-26 in lines 70-74. The original hypothesis is that vascular dysregulation is the inciting event in AD pathogenesis. Lines 74-82 clarify revised versions of the vascular hypothesis which postulate 2 hits: first vascular injury (hit 1) then amyloid aggregation (hit 2) which further aggravates vascular injury.

  1. Line 92, introduce APOE4 and a brief description about how APOE4 is involved in AD pathology, and it is strongest risk factor for AD/dementia development. 

Response: This has now been included in lines 97-102. I describe APOE4 as a genetic risk factor for AD and the mechanisms by which it is implicated in AD pathogenesis.

  1. Lines 96-103, it be beneficial to include studies and mechanisms about how LDL, homocysteine are linked to AD. 

Response: This has now been addressed in lines 113-115 for LDL and lines 125-128 for homocysteine.

  1. Line 154-157, provide an original reference rather than a review paper. 

Response: References 10, 16-18 are original papers and now included in that statement (on line 168).

  1. Regarding lines 187-188, it be beneficial to provide the list of 45 AD risk and their site of expression. 

Response: We thank the reviewer for this suggestion. This has been included in a new table (Table 1) on pages 5-6.

  1. Reference(s) required in line 230. How are astrocytes involved in Aβ synthesis? Is it via primarily via APOE4?

Response: Reference 94 has been added for lines 264-270. This section has been considerably revised to clarify that astrocytes are involved in Aβ metabolism by both APOE4-dependent and -independent processes.

  1. Introduce β- and γ-secretase and their function in line 281. 

Response: I have now clarified the role of these enzymes in the APP amyloidogenic pathway on lines 337-339.

  1. Reference(s) required in lines 309-311. 

Response: References 129-130 have now been added on lines 365-366.

  1. Figure legend missing in both figures. 

Response: Legends for both figures were included with the original submission but inadvertently omitted during formatting for the journal template. I have now included the legends for all 3 original figures in the main manuscript file (a new Figure 3 has been added with the revisions).

  1. What is the role of VDAC1?

Response: Lines 439-440 clarify that voltage gated anion channel 1 is a gamma secretase activator.

  1. Reference(s) required in lines 414-416, 429-430, and 436-438. 

Response: References 170-171 has been added to lines 484-485. References 168-169 are cited in line 479-482. Lines 498-499 have reference 176.

  1. Information in section 5.6 is linked to neuronal lipid raft or endothelial.

Response: I thank the reviewer for this clarification. I have made minor revisions in that section to clarify that the first paragraph is mostly about neuronal lipid rafts and the second paragraph is about endothelial lipid rafts.

  1. Introduce GLUT1

Response: This has been added to line 617.

  1. Is Figure 2 taken from previously published manuscript or newly generated. Provide details in figure legend. 

Response: All figures in this paper are original figures. Figures 1 and 3 were created by BioRender and Figure 2 by STRING. The legends were included with the original submission but were inadvertently omitted during formatting for the journal template. All details related to Fig 2 are now shown in its legend.

Reviewer 2 Report

3.  This section might confuse readers understanding the distinction between what the author terms macrovascular and microvascular.  The first sentence “Macrovascular disease, including subcortical white matter lesions and lacunes…” is somewhat discrepant with text in the same paragraph equating “small vessel disease” with subcortical lacunes and white matter lesions.  Maybe a segue could mention that microvascular injury extends beyond the arteriolar level. 

5.3  It’s not clear to me what the last sentence (403-406) has to do with endoplasmic reticulum stress. 

Figure 2 doesn’t convey a lot of meaning on its own.  The relevant text (554-561) mentions interactions between endothelial proteins and neuronal proteins (semaphorin 3A, neuropilin/plexin-A1, Nr-CAM).  The figure includes other elements this reader had to look up (CDH5 is VE-Cadherin) to relate the depicted relationship between endothelial proteins alluded to in earlier sections.  Maybe a figure legend might clarify understanding.   

Author Response

Response: I thank the reviewer for this positive feedback and thoughtful review of this manuscript.

I have addressed all reviewer comments here and in the revised document (marked in blue).

  1. This section might confuse readers understanding the distinction between what the author terms macrovascular and microvascular.  The first sentence “Macrovascular disease, including subcortical white matter lesions and lacunes…” is somewhat discrepant with text in the same paragraph equating “small vessel disease” with subcortical lacunes and white matter lesions.  Maybe a segue could mention that microvascular injury extends beyond the arteriolar level. 

Response: I thank the reviewer for bringing this to my attention. I have clarified the terms in the introduction lines 42 and 56 to state that “micro-vascular” in this manuscript refers to the level of the capillaries.

5.3  It’s not clear to me what the last sentence (403-406) has to do with endoplasmic reticulum stress. 

Response: I thank the reviewer for bringing this to my attention. These statements were supposed to be separate from that section (i.e., a new paragraph) but appear to have been misplaced during formatting. These statements were included to introduce mechanisms by which endothelial cells contribute to AD pathogenesis, whereas the sections prior to that covered converging mechanism that were shared for both endothelial injury and hypoxia. I have now separated this introduction from the preceding ER Stress section.

Figure 2 doesn’t convey a lot of meaning on its own.  The relevant text (554-561) mentions interactions between endothelial proteins and neuronal proteins (semaphorin 3A, neuropilin/plexin-A1, Nr-CAM).  The figure includes other elements this reader had to look up (CDH5 is VE-Cadherin) to relate the depicted relationship between endothelial proteins alluded to in earlier sections.  Maybe a figure legend might clarify understanding.   

Response: I thank the reviewer for bringing this to my attention. The legends were included with the original submission but were inadvertently omitted during formatting for the journal template. All details related to Fig 2 are now shown in its legend.

Reviewer 3 Report

Well written and concise review. No issues noted. 

Author Response

Response: I would like to thank the reviewer for their positive feedback.

Reviewer 4 Report

This challenging review highlight the role of dysfunctional endothelium in neurodegeneration in AD.

-          The following sentence “30 out of 45 AD risk genes are expressed in endothelial cells, and several of these had their highest expression levels in endothelial structures“ should be presented more in detail. A table might be helpful.

-          In paragraph 5, BDNF is not mentioned. It is an important issue since brain microvascular endothelial cells are an important source of BDNF in the CNS.

-          Add a legend to the figure.

-          Angiogenesis is impaired in all aging tissues because of the decrease of angiogenic factors. I feel that too much attention is devoted by the author to an impairment of angiogenesis in AD. It is more likely that there is an impairment of endothelial function rather than an impairment of angiogenesis implicated in reducing capillary density.

-          In the figure capillary degeneration results also from endothelial dysfunction.  A two-way arrow is more appropriate. I suggest to eliminate “angiogenesis impairment”, so that oxidative stress is the cause of neuronal injury.

-          “Aβ toxicity and low MEOX-2 levels” can not be defined anti-angiogenic factors.

Author Response

This challenging review highlight the role of dysfunctional endothelium in neurodegeneration in AD.

The following sentence “30 out of 45 AD risk genes are expressed in endothelial cells, and several of these had their highest expression levels in endothelial structures“ should be presented more in detail. A table might be helpful.

Response: I thank the reviewer for this thoughtful suggestion. I have now included Table 1 on pages 5-6 which lists the 45 genes and their expression profiles.

In paragraph 5, BDNF is not mentioned. It is an important issue since brain microvascular endothelial cells are an important source of BDNF in the CNS.

Response: I thank the reviewer for this thoughtful suggestion. I have now added a new subsection describing the role of endothelial BDNF in neurogenesis (Lines 671-703). I have also added a new Figure 3 which summarizes the pathways by which endothelial BDNF is processed in neurons, astrocytes, and endothelium.

Add a legend to the figure.

Response: I thank the reviewer for bringing this to my attention. Legends for both figures were included with the original submission but were inadvertently omitted during formatting for the journal template. I have now included the legends for all 3 figures in the main manuscript file.

Angiogenesis is impaired in all aging tissues because of the decrease of angiogenic factors. I feel that too much attention is devoted by the author to an impairment of angiogenesis in AD. It is more likely that there is an impairment of endothelial function rather than an impairment of angiogenesis implicated in reducing capillary density. In the figure capillary degeneration results also from endothelial dysfunction.  A two-way arrow is more appropriate. I suggest to eliminate “angiogenesis impairment”, so that oxidative stress is the cause of neuronal injury.

Response: I thank the reviewer for their suggestions. I discuss angiogenesis as an endothelial function that is impaired in AD and one of the mechanisms by which Aβ or hypoxia induce endothelial injury. Angiogenesis is also (heavily) addressed by many papers in the field (Grammas papers ref 14, and Paris 201) including the importance of impaired VEGF signaling. That being said, in response to the reviewer’s suggestion, I have removed "impaired angiogenesis" from Figure 1 and added “impaired neurogenesis” instead in the revised Figure 1. I have also revised the arrows in Fig 1 (that connect capillary and endothelial injury) to 2-way arrows. I agree with the reviewer this is more descriptive and thank them for their suggestions.

      “Aβ toxicity and low MEOX-2 levels” can not be defined anti-angiogenic factors.

Response: I have removed the term “anti-angiogenic” and rephrased that sentence on line 548-549.

Reviewer 5 Report

The review paper by Tarawneh summarizes the literature on the issue of AD pathology involving microvasculature dysfunctions. The author apparently surveyed large number of papers and outlined the potential role of cerebral microvessels, especially endothelial, on the pathogenesis of AD in its early phase of development. The discussion on the hypoperfusion and dementia, the hypoxia and endothelial dysfunction (part 5) is very informative and well written. However, there are a few issues the author could address to improve the review.

1. In the title the author asked the question “Is Alzheimer disease primarily an endotheliopathy?” It is certainly an interesting one, but the author did not give an answer with definitive evidence to indicate yes or no. Rather, the literature review gives the impression that APP or Tau aggression and endothelial dysfunction actually are interwoven and difficult to set it clear as which one plays a dominant role in early phase of AD pathology. Therefore, the question in title is somewhat an over-statement. The author may want to turn it down as no clear evidence in the whole paper supports even the hint of AD is “primarily an endotheliopathy”.

2. The reference issue. It could be more informative for readers to see the original paper when a topic is cited. For example, the evidence that there is endothelial and capillary degeneration but without macrovascular pathology in AD brain is better presented with the original reports, not other review papers such as ref#4 and 5 (line 32-33). Again, the author states that the vascular hypothesis of AD raised 30 years ago but the reference is another review of 2022 (ref 11, line 72). What are the original papers published 30 years ago? Please cite these old papers, not another review paper. There is no mention of homocysteine in ref #15 even though the authors cited for the role of homocysteine in AD pathology (line 111-113).

3. In this single author review paper, should the author use I, instead of we, our? (line 48, 54).

4. Line 159-161, it is stated that “Animal studies have shown that significant alterations to brain microvasculature are evident in young AD transgenic mice and precede brain amyloid pathology8,51,52”. It is an interesting observation, but it would be more impactful with a brief mechanistic explanation why capillary abnormalities occurred prior to the AD pathology in these mice.

Author Response

The review paper by Tarawneh summarizes the literature on the issue of AD pathology involving microvasculature dysfunctions. The author apparently surveyed large number of papers and outlined the potential role of cerebral microvessels, especially endothelial, on the pathogenesis of AD in its early phase of development. The discussion on the hypoperfusion and dementia, the hypoxia and endothelial dysfunction (part 5) is very informative and well written.

Response: I thank the reviewer for their positive feedback and thoughtful review of our manuscript.

However, there are a few issues the author could address to improve the review.

  1. In the title the author asked the question “Is Alzheimer disease primarily an endotheliopathy?” It is certainly an interesting one, but the author did not give an answer with definitive evidence to indicate yes or no. Rather, the literature review gives the impression that APP or Tau aggression and endothelial dysfunction actually are interwoven and difficult to set it clear as which one plays a dominant role in early phase of AD pathology. Therefore, the question in title is somewhat an over-statement. The author may want to turn it down as no clear evidence in the whole paper supports even the hint of AD is “primarily an endotheliopathy”.

Response: I thank the reviewer for their thoughtful comment. This review summarizes the evidence (from animal and clinical studies) and mechanisms by which endothelial injury contributes to AD pathogenesis including amyloid and tau aggregation which collectively support the vascular hypothesis: endothelial injury and reduced CBF are the inciting events in AD pathogenesis which trigger the AD cascade. Support for this also comes from animal studies which show that endothelial injury precedes amyloid aggregation and longitudinal clinical studies from the ADNI cohorts (with >20 years of follow-up). These clinical studies generated new predictive disease models which clearly show that vascular flow changes precede amyloid and tau aggregation (Itturia-Medina et al, Ref 19). Therefore, most of this review is focused on presenting evidence that supports that AD may in fact primarily be an endotheliopathy in its origin. However, more data is needed to support this hypothesis. I clarify these points on lines 714-721.

  1. The reference issue. It could be more informative for readers to see the original paper when a topic is cited. For example, the evidence that there is endothelial and capillary degeneration but without macrovascular pathology in AD brain is better presented with the original reports, not other review papers such as ref#4 and 5 (line 32-33). Again, the author states that the vascular hypothesis of AD raised 30 years ago but the reference is another review of 2022 (ref 11, line 72). What are the original papers published 30 years ago? Please cite these old papers, not another review paper. There is no mention of homocysteine in ref #15 even though the authors cited for the role of homocysteine in AD pathology (line 111-113).

Responses: I thank the reviewer for their suggestion. References 4-7 have been added on lines 33-34. References 23-26 are included on line 74 for the vascular hypothesis.

I thank the reviewer for bringing previous Ref 15 to my attention. I have now replaced this with another references 54-56.

  1. In this single author review paper, should the author use I, instead of we, our? (line 48, 54).

Response: This is commonly done in the literature. However, I will defer to the editors' discretion. 

  1. Line 159-161, it is stated that “Animal studies have shown that significant alterations to brain microvasculature are evident in young AD transgenic mice and precede brain amyloid pathology8,51,52”. It is an interesting observation, but it would be more impactful with a brief mechanistic explanation why capillary abnormalities occurred prior to the AD pathology in these mice.

Response: I thank the reviewer for their suggestion. I have added a paragraph lines 209-217 to describe possible mechanisms underlying endothelial injury reported in these studies and clarify that further studies are warranted to better understand these findings.